# Vaccine Quality Is a Key Factor to Determine Thermal Stability of Commercial Newcastle Disease (ND)Vaccines

**DOI:** 10.3390/vaccines9040363

**Published:** 2021-04-09

**Authors:** Nabila Osman, Danny Goovaerts, Serageldeen Sultan, Jeremy Salt, Christian Grund

**Affiliations:** 1Department of Poultry Diseases, Faculty of Veterinary Medicine, South Valley University, Qena 83523, Egypt; nabila.osman@vet.svu.edu.eg; 2DGVAC Consulting, 2460 Antwerp, Belgium; Danny.Goovaerts@galvmed.org; 3GALVmed, Edinburgh EH26 0PZ, UK; Jeremy.Salt@galvmed.org; 4Department of Microbiology, Virology Division, Faculty of Veterinary Medicine, South Valley University, Qena 83523, Egypt; sultanserageldeen@gmail.com; 5Institute of Diagnostic Virology, Friedrich-Loeffler-Institut, Edinburgh EH26 0PZ, UK

**Keywords:** heat stability, Newcastle disease virus, paramyxovirus, vaccine quality, vaccine stability

## Abstract

Vaccination against Newcastle disease (ND), a devastating viral disease of chickens, is often hampered by thermal inactivation of the live vaccines, in particular in tropical and hot climate conditions. In the past, “thermostable” vaccine strains (I-2) were proposed to overcome this problem but previous comparative studies did not include formulation-specific factors of commercial vaccines. In the current study, we aimed to verify the superior thermal stability of commercially formulated I-2 strains by comparing six commercially available ND vaccines. Subjected to 37 °C as lyophilized preparations, two vaccines containing I-2 strains were more sensitive to inactivation than a third I-2 vaccine or compared to three other vaccines based on different ND strains. However, reconstitution strains proved to have a comparable tenacity. Interestingly, all vaccines still retained a sufficient virus dose for protection (10^6^ EID_50_) after 1 day at 37 °C. These results suggest that there are specific factors that influence thermal stability beyond the strain-specific characteristics. Exposing ND vaccines to elevated temperatures of 51 and 61 °C demonstrated that inactivation of all dissolved vaccines including I-2 vaccine strains occurred within 2 to 4 h. The results revealed important differences among the vaccines and emphasize the importance of the quality of a certain vaccine preparation rather than the strain it contains. These data highlight that regardless of the ND strain used for vaccine preparation, the appropriate cold chain is mandatory for keeping live ND vaccines efficiency in hot climates.

## 1. Introduction

Newcastle disease (ND) is one of the most devastating endemic viral diseases of poultry in many countries worldwide [1,2]. The economic impact of this is severe—for example, with estimated losses of USD 288 million annually in Bangladesh [3], USD 23 million in Nigeria [4] and USD 162 million in the United States of America [5]. The disease is caused by *avian orthoavulavirus 1*(AOAV-1, ND virus (NDV)) within the subfamily *Avulavirinae* of the family *Paramyxoviridae* [6]. Depending on the virulence of the ND virus (NDV), infections can cause a wide spectrum of clinical signs ranging from asymptomatic (apathogenic, lentogenic pathotype) or mild clinical respiratory signs or a drop of egg production (mesogenic pathotype) to up to 100% mortality (velogenic pathotype) [7]. One of the hallmarks of ND prevention was the discovery of lentogenic NDVs in America [8,9]. The subsequent development of the vaccine strains La Sota and B1 with derivatives such asclone30 are the basis for most ND vaccines. Some strains of NDV isolated in Australia between 1967 and 1978 were also found to be avirulent [10], and the V4 strain, which was claimed to have enhanced thermostability, was used to develop live ND vaccines with the I-2 strain as its variant.

Today, ND vaccination is applied worldwide and is effective in reducing the impact of the disease. Particularly for smallholder chicken productivity in the developing world, backyard chicken ND vaccination is an important tool for poultry health [11,12]. However, easy thermal inactivation of the live virus vaccine in hot climates and distant regions can affect its efficacy [13,14,15]. Akin to most live vaccines, paramyxovirus-based ND vaccines are heat labile and require a cold chain to preserve the quality of vaccines during transport and storage. Reliability of the cold chain is a challenge and temperature excursions outside the optimal temperature range are frequently observed during transport and storage [16,17]. Inappropriate equipment, human errors and power shortages are important causes of cold chain breaking [18,19]. It is estimated that roughly 50% of all lyophilized vaccines are discarded annually, and poor thermostability is an important contributing factor in this issue [20].

To tackle this problem, several so-called thermostable strains have been described, including V4, TS09-C, and I-2 [21]. Phylogenetically, all these strains belong, as the Ulster strain does, to class 1 viruses, genotype I. The I-2 strain was selected based on the recovery of infectious virus after exposure to 56 °C for 3 h [22], while strain TS09-C was developed by serial passages of strain V4 in BHK-21 cells [23]. Vaccines prepared from some of these strains are thought to be safe, thermostable, immunogenic, able to spread between chickens, and suitable for delivery on food [24,25].

What determines the stability of a certain vaccine might depend on intrinsic properties of the strain, but other factors such as manufacturing and lyophilization processes can have an equally important impact. Whereas studies on properties of specific strains in a standardized formulation are available [26,27], data on the influence of the galenics for thermostability of ND vaccines are missing so far. To address this question, we tested a number of commercially available freeze-dried preparations, comparing three conventional and three so-called thermostable vaccines containing the I-2 strain.

Studying commercially available ND vaccines will elucidate further possible reasons for vaccination failure, addressing the impact of manufacturing aspects of freeze-dried vaccines. The results are important in the development of successful vaccination strategies for hot climates based on appropriate supply chains, the selection of adequate ND vaccine strains and application routes to ensure that birds will be vaccinated with sufficiently protective doses of a live ND vaccine virus.

## 2. Material and Methods

For this investigation, six commercial vaccines against ND were tested, four of them claiming enhanced thermostability (A, D–F) and two not (B, C), (Table 1). The vaccine shipment was organized by GALVmed and reached the FLI in good conditions. All vaccines were used well within their shelf lives. Upon arrival, vaccines were stored at 4 °C. Exposure to elevated temperatures (37, 41, 51 and 61 °C) was carried out in a water bath for the indicated periods, using either the original lyophilized material or vaccine vial diluted in 2 mL sterile double distilled water (ddH_2_O). Temperatures were chosen based on the temperature in an incubator for cell cultures (37 °C), similar to the body temperature of chickens (41 °C), and wereincreasedby10 °C (i.e., 51to 61 °C). After the indicated exposure times, all vials were stored at −70 °C until they were tested for infectivity and hemagglutinating activity. As a reference, lyophilized vaccine vials were kept at 4 °C and tested on days 0, 7 and 21 after the start of the experiment.

### 2.1. Determination of Infectivity

Infectivity was determined for vaccine vials diluted in 2 mLddH_2_O and subsequently inoculated either on LMH cell culture (ATCC^®^ CRL-2117™; [28]) or using specific pathogen free (SPF) chicken eggs (VALO BioMedia, Germany) following standard procedures [29]. In short, tenfold serial dilutions were prepared in cell culture medium (DMEM) and 0.2 mL of each dilution was inoculated in the allantoic sac of five 10-day-old embryonated SPF chicken eggs. After 5 days, eggs were chilled at 4 °C for 2 h and amnion-allantoic fluid (AAF) was harvested and subsequently tested for hemagglutinating (HA) activity. Eggs with HA-positive AAF but negative for bacteria were considered AOAV-1 infected.

For determination of the infectious dose in cell culture, LMH cells were freshly transferred to 96-well plates in medium supplemented with TCPK-Trypsin (2 µg/mL) but without fetal calf serum. The addition of exogenous protease enabled multicycle replication of lentogenic vaccine strains with induction of multinucleated giant cells and subsequent cell necrosis. For each time point, two independent tenfold virus dilutions in cell culture medium were prepared and 2 × 4 replicates for each dilution with 50 µL each were transferred to the cells. Three days after infection, wells were investigated by microscopy for the induced cytopathic effect.

Egg infectious dose 50 (EID_50_) and tissue culture infectious dose 50 (TCID50/mL) were calculated according to the method by Reed and Muench [30]. Values were normalized to the vaccine dose within vaccine vials.

### 2.2. Hemagglutination Assay

All samples from different time/temperature intervals were investigated in parallel to infectivity titers for hemagglutination according to standard protocols [29].

## 3. Results

Investigation of vaccines stored at 4 °C over a period of 21 days resulted in virus titers well above 10^6^ EID_50_ per dose at any time (Figure 1A). Fluctuation of the virus titer was within a limit of one dilution (log10) and can be considered as an indication for the variations of the biological titration system in embryonated chicken eggs. Virus titers measured by cell culture were lower, but clearly support the notion of stability of lyophilized ND vaccines at 4 °C (Figure 1B). Results of the HA units (HAUs) were in line with titration as HAUs remained stable for each of the vaccines over the entire observation period (Appendix A).

At 37 °C, lyophilized vaccines again started out with virus titers well above 10^6^ EID_50_per dose (9.5 × 10^6^–2.7 × 10^8^ EID_50_/dose) (Figure 2A). Titration on cells resulted in lower virus titers (3 × 10^5^–9.7 × 10^6^ TCID_50_/dose)—on average, a 36-fold difference (5–85-fold) compared to titers obtained from egg culture was observed (Figure 2B). The ratio between EID_50_ and TCID_50_ remained within this range for the subsequent time points, with 14-, 28- and 11-fold higher EID_50_values for samples collected 10, 14 and 21 day safter exposure (dpe) at 37 °C, respectively. Over this 3-week period, EID_50_ remained above 10^5^ for 4 out of the 6 vaccines tested (1.3 × 10^5^–1.3 × 10^6^). Infectivity of two out of three vaccines containing the I-2 strain (Figure 2A, vaccine D,F) dropped dramatically, with vaccine F becoming negative by day 14 after exposure to 37 °C and vaccine D only having minimal residual virus titers in samples from day 10 to 21 (1.4–2 × 10^1^ EID_50_). The decay of infectivity became apparent on days 5 (vaccine F) and 10 (vaccine D) (Figure 2A). In contrast, HAactivity remained at the same level for the entire observation period (Appendix A).

Considering testing stability for dissolved vaccines (Figure 2C,D), it was remarkable that at the end of the 4-day observation period the virus titers of 4 out of 6 viruses were above 10^5^ EID_50_ (1.1 × 10^5^–9.5 × 10^6^ EID_50_)(Figure 2C). Vaccines B and F still had virus titers of above 10^6^ EID_50_, an amount that is indicated as vaccine dose (2.7 × 10^6^ EID_50_ and 9.5 × 10^5^ EID_50_,respectively), while vaccines A, D and E contained 2.5 × 10^5^, 1.1 × 10^5^ and 5.8 × 10^4^ EID_50_ per dose, respectively. Loss of infectivity higher than 10^4^ EID_50_was observed only for vaccine C, resulting in a final virus titer of 5.5 × 10^2^ EID_50_ per dose. The same trend was evident for the samples tested by cell culture (Figure 2D). Again, after vaccine dissolution, HA activity remained at the same level for the entire observation period (Appendix A).

Comparing the data from both experimental settings, it became apparent that the displayed loss of infectivity for vaccines D and F was particularly evident with lyophilized vaccine preparations, while losses in solution were less pronounced. To verify these results, fresh virus stocks from all vaccine strains were prepared by propagating viruses in the allantoic cavity of embryonated SPF chicken eggs. The obtained AAF was frozen at -70 °C as virus stocks. Virus titers were determined from one aliquot and ranged between 1.3 × 10^8^ TCID_50_(Vaccine A) and 1.3 × 10^9^ TCID_50_ (Vaccine D).

To test strain thermostability, fresh aliquots of the prepared virus stocks were diluted with DMEM medium to give a final titer of 10^8^ TCID_50_and were subsequently exposed to a temperature of 37 °C in a water bath. Comparable to the previous experiment with the original vaccine, loss of infectivity remained moderate for the first day, from 2.0 × 10^7^ to 1.3 × 10^8^ TCID_50_, representing 16–135% residual infectivity (Figure 2F). An exception to this trend is virus stock A, with a drop in the virus titer to 5.0 × 10^6^ TCID_50_, representing 0.97% residual infectivity. However, on day 3, the decay of infectivity slowed down for virus stock A, resembling the results of four other virus strains, with a virus titer of 1.3 × 10^6^ to 1.0 × 10^7^ TCID_50_representing 0.3–5.8% residual infectivity (Figure 2F). At this point in time, virus stock B showed the most prominent decline in virus titer, dropping to 6.3 × 10^4^ TCID_50_, representing 0.068% residual infectivity on day 3 and becoming negative on day 5. At day 7, temperature exposure infectivity was present in all other vaccine virus stocks, and at the end of the experiment on day 10 residual infectivity was detectable only in viral stocks from vaccines D, E and A with virus titers of 3.7 × 10^0^, 1.4 × 10^2^, and 1.6 × 10^2^ TCID_50_, respectively.

The results obtained for a temperature of 37 °C demonstrate that some NDV vaccines, irrespective of the strains tested, are remarkably stable at a temperature close to the physiological environment encountered during virus replication. However, in hot climates more elevated temperatures might also be relevant with live NDV vaccination. Therefore, the effect of exposure of vaccines and vaccine virus stocks to 51 and 61 °C was investigated. In addition to the original vaccine B, viral vaccine stocks were tested (Figure 3). It became evident that lyophilized vaccine could withstand 41 °C and even 51 °C for one day with infectivity of 1.5 × 10^5^ TCID_50_ per dose (Figure 3A). At day three at 51 °C, the virus titer dropped to 2.3 × 10^3^ TCID_50_ and after that only residual infectivity was left. At 61 °C infectivity declined more rapidly but was still present on days 1 (1.3 × 10^2^ TCID_50_) and 3 (1.3 × 10^1^ TCID_50_). Dissolved vaccine could not withstand elevated temperatures: at the first time point tested, i.e., 6 h of exposure, infectivity was below the detection limit (Figure 3B). In subsequent experiments, where different fresh stocks of vaccine strains were tested at 51 (Figure 3C) and 61 °C (Figure 3D), rapid decay was verified. After 2 h exposure at 51 °C, residual virus titers were between 3.7 × 10^1^ TCID_50_ (vaccine virus B) and 1.5 × 10^4^ TCID_50_ (vaccine virus F). After 4 h, residual infectivity was only detected for vaccine viruses D and E (3.7 × 10^1^ TCID_50_). At 61 °C, inactivation was complete after 2 h, irrespective of the strain used in the vaccines.

## 4. Discussion

Vaccines are an effective tool in disease prevention, but the efficacy of live vaccines is hampered by thermal inactivation, particularlyin hot climates [11]. To minimize temperature-induced degradation, virus strains such as I-2 that are considered thermostable were proposed to be most suitable for ND vaccination in rural areas in hot climates [31]. Studies on this subject have mainly focused on the use of different vaccine strains, but the intrinsic quality of vaccine preparations and the importance of freeze-dry technology have not been consideredequally. Here, we report on six commercial products that represent four vaccines claiming increased thermostability, three based on the use of the I-2 strain (vaccines D, E and F)and one containing an improved freeze-dry process with the La Sota strain(vaccine A). Two vaccines not claiming enhanced thermostabilityare based on La Sota derivatives (vaccines B and C). The results revealed that all vaccines contained viral titers above 10^6^ EID_50_ per dose in the original material and were stable as lyophilized preparation at 4 °C. However, after incubation at 37 °C infectivity dropped dramatically for two lyophilized vaccines containing I-2 (vaccines D and F), whereas the other four vaccines retained a virus titer above 10^5^ EID_50_ until the end of the investigation period of 21 dpe. Our results revealed that the La Sota vaccine (vaccine A) and vaccines based on its derivatives (vaccines B and C) are relatively stable at an ambient temperature of 37 °C. For La Sota (vaccine A), this is in agreement with previous work [31], but for clone30 (vaccine B) our data contradict previous findings. While in Boumartet al.’s [31] study complete loss of infectivity of the freeze-dried clone30 was observed when tested at 14 dpe, our study results revealed a loss of only less than 1 log10 over this period.

A possible explanation might be that in contrast to our study, Boumart and colleagues [31] did not test commercially ready-to-use vaccines but freeze-dried AAF preparations using the same stabilizer for all strains, thus disregarding the impact of quality and manufacturing aspects of the individual manufacturer. This striking difference between the two studies points to the influence of manufacturing factors, which is in agreement with earlier investigations, comparing seven vaccines containing either La Sota or Hitchner B1 strains [32]. Individual manufacturers could potentially have selected the most optimal stabilizer combinations and freeze-dry conditions depending on the vaccine strain and as such have considerably improved robustness of vaccine preparation. Such differences among products may include vaccine formulation, stabilizers, methods of vaccine preparation, and residual moisture of the freeze-dried pellet, factors that can considerably influence quality on the level of vaccine batches [20,33,34].

To elaborate further whether strain differences or manufacturing aspects were the basis of the observed differences between individual vaccines, fresh AAF stocks were prepared in the laboratory and subjected to elevated temperatures. A decrease in infectivity after 3 days was comparable to the dissolved commercial vaccines with a drop of two log10 TCID_50_. However, in this experiment the clone30 preparation (vaccine B) was the most sensitive to heat treatment. This became even more pronounced after five dpe. At this point in time no residual infectivity was left in the clone30 stock (vaccine B), while all other viruses still yielded considerable infectivity. In conjunction with the first set of experiments, this further supports the notion that thermostability of a given vaccine does not depend solely on the used vaccine strain, but that the overall excellence of the manufacturing process can be as or even more important for the quality of the vaccine. This was also clearly demonstrated by the important differences found among the three vaccines, all using the I-2 strain but which are clearly different with respect to thermostability. It is striking that two out of three vaccines based on the I-2 strain and claiming enhanced thermostability were considerably less stable at elevated temperatures compared to the other vaccines. Moreover, none of the vaccines claimed to have enhanced thermostability performed better compared to the vaccines containing the clone30(vaccine B) and Cl/79 (vaccine C) strains, for which no thermostability properties have been claimed.

## 5. Conclusions

A specific definition and criteria for thermostability (time, temperature, acceptable losses) in the literature and in regulatory requirements are currently lacking. It is clear from our data that for the benefit of the user strict criteria are urgently needed and need to be properly controlled in case certain manufacturers want to make such claims for commercial reasons. However, our results also clearly demonstrate that enhanced thermostability or any other characteristic of a certain vaccine should be evaluated as a property of the product and should not solely rely on a part of the content of such a product. In evaluating the quality of a vaccine, one cannot only rely on a single ingredient, e.g., the strain, but every individual aspect will finally have its impact on the overall outcome. The LMH cell culture appears to be a suitable test system for such a characterization process. Compared to embryonated egg culture, results in both systems gave very comparable results and were well correlated (Appendix A). However, comparison to the HA results clearly show that measuring HA activity is insufficient to monitor the biological stability of a vaccine.

From a practical point of view, the rapid decay of infectivity at 51and 61 °C indicates that all vaccines tested are prone to inactivation at elevated temperatures. This is in line with earlier observations [35] and should be considered for application of live ND vaccines. Not only in hot climates but also on hot days in moderate climates, water pipes and reservoirs might become warmed up and accelerate degradation of the ND vaccine. Thus, in addition to errors due to inappropriate storage temperature, human mishandling or defective transport equipment [16,17], inappropriate application may account for vaccination failure. In order to ensure the application of an appropriate vaccine dose, maintaining the cold chain for live ND vaccines is necessary.

## Figures and Tables

**Figure 1 vaccines-09-00363-f001:**
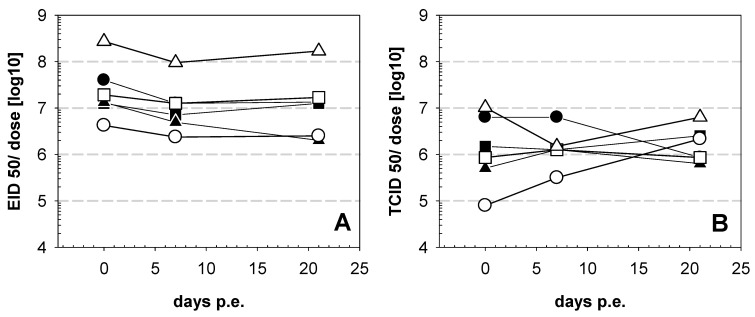
Stability of commercial Newcastle disease (ND) vaccines stored at 4 °C. Lyophilized original vaccine vials were stored at 4 °C and at indicated time points after the start of the experiments vials were investigated for infectivity on embryonatedspecific pathogen free (SPF) chicken eggs (**A**) and on LMHcells (**B**). Symbols represent vaccine A (●), vaccine B (■), vaccine C (▲), vaccine D (○) vaccine E (□), and vaccine F (∆).

**Figure 2 vaccines-09-00363-f002:**
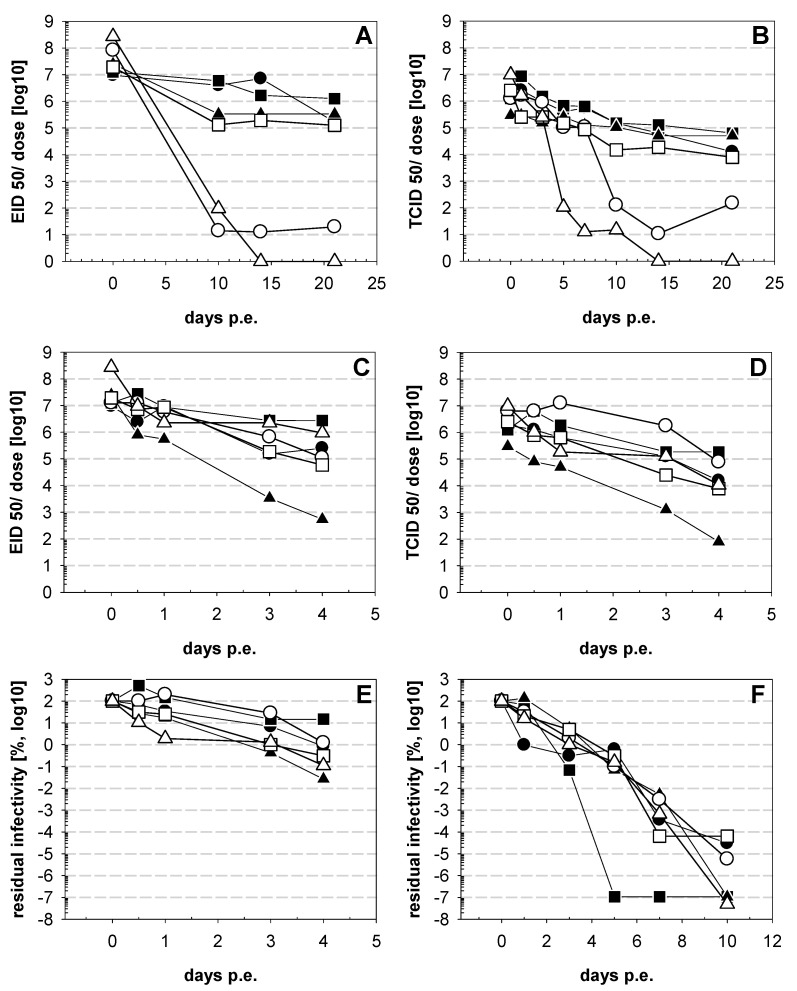
Stability of ND vaccine at 37 °C. ND vaccines were exposed to 37 °C in a water bath for indicated days (dpe). In the first experiments the vaccines were incubated as lyophilized materials in the original package and tested for residual infectivity using embryonatedSPF chicken eggs (**A**) or LMHcells (**B**). The second experiment tested vaccines dissolved in 2 mL ddH_2_O and determined residual infectivity with embryonatedSPF chicken eggs(**C**) or LMHcells (**D**). The infectivity is normalized per dose of the original package. In a third set of experiments amnion-allantoic fluid (AAF) from virus stocks generated in the laboratory in embryonated SPF chicken eggs were used and tested for residual infectivity on LMH cells (**F**). To normalize results, residual infectivity is given as % of the original virus stock. For comparison, data from dissolved vaccines (C) are included as % residual infectivity (**E**). Symbols represent vaccine A (●), vaccine B (■), vaccine C (▲), vaccine D (○) vaccine E (□), and vaccine F (∆).

**Figure 3 vaccines-09-00363-f003:**
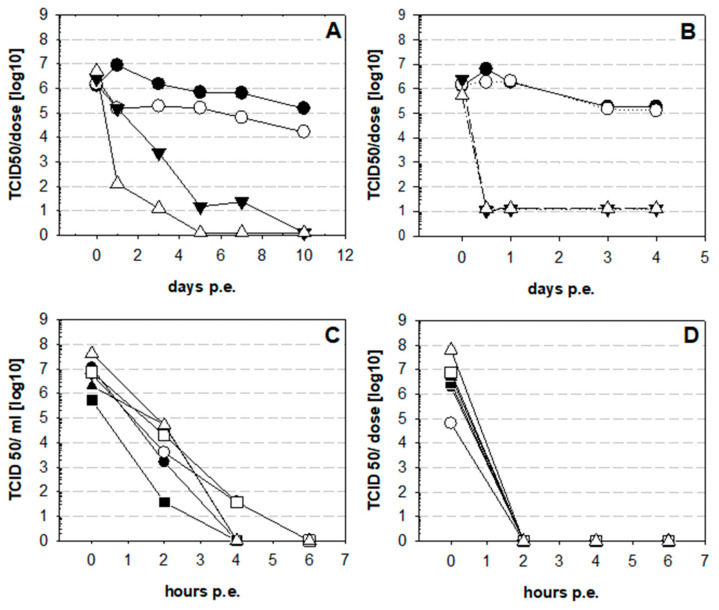
Stability of NDvaccines at elevated temperatures.Vaccine B was tested at elevated temperatures as lyophilized preparation in the original tubes (**A**) or dissolved (**B**) at 37 (●), 41 (○), 51 (▼), 61 °C (∆). Viral titer was normalized to virus dose in the vaccine. In addition, prepared AAF stocks of all six vaccines were tested after exposure to 51 (**C**) or at 61 °C (**D**). Symbols represent vaccine virus A (●), vaccine virus B (■), vaccine virus C (▲), vaccine virus D (○) vaccine virus E (□), and vaccine virus F (∆).

**Table 1 vaccines-09-00363-t001:** Tested live Newcastle disease vaccines.

Code	Vaccine		Strain	Dose/Vial	Vol. ^1^	EID ^2^	TCID ^2^	HAU ^3^	EID/TCID	EID50/HAU
A	La Sota thermostable ND	Hester	La Sota	100	2 mL	4.7 × 10^6^	9.3 × 10^5^	128	5.1	3.7 × 10^6^
B	ND clone 30	MSD AH	clone 30	1000	2 mL	6.3 × 10^6^	6.3 × 10^6^	128	10.0	4.9 × 10^7^
C	Hipraviar CLON	Hipra	CI/79	5000	2 mL	1.3 × 10^7^	1.3 × 10^5^	512	84.8	1.2 × 10^8^
D	ND-Kukustar	Brentec	I-2	500	1 mL	8.3 × 10^7^	6.3 × 10^8^	256	65.8	1.6 × 10^8^
E	Avivax ND I-2 lyophilisé	MCI Santé Animale	I-2	100	2 mL	9.6 × 10^6^	1.3 × 10^6^	512	7.6	1.9 × 10^6^
F	Avivax I-2	Kevevapi	I-2	100	2 mL	1.4 × 10^8^	4.9 × 10^6^	64	28	2.1 × 10^6^

^1^ Volume used to dissolve the vial; ^2^ infection dose 50 per vaccine dose; ^3^ hemagglutination units in dissolved preparations.

## Data Availability

The datasets during and/or analyzed during the current study available from the corresponding author on reasonable request.

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
