# Peer review of "Vaccine Quality Is a Key Factor to Determine Thermal Stability of Commercial Newcastle Disease (ND)Vaccines"

_vaccines, 2021, doi:10.3390/vaccines9040363_

Round 1

Reviewer 1 Report

In the abstract, could you consider rewriting the following sentence? This reviewer was unable to understand the idea on this sentence: “The data indicate preparation that specific factors may influence thermal stability rather than strain specific characteristics.” It can be unerstood only after finishing reading the manuscript.

In my opinion the abstract lacks clarity at the end. By reading the abstract I cannot realize at its full extent what the conclusions are based on the main findings. Please revise it. Unlike the abstract, the Introduction provides more elements and clarity.

Could you please confirm the residual titers in Page 5:  “While at day 7 after temperature exposure infectivity was present in all other vaccine virus stocks, at the end of the experiment on day 10, residual infectivity was detectable only in viral stocks from vaccine D, E and A with viral-titer of 3.7x100, 1.4x102, and 1.6x102 TCID50 respectively”.

Figure 2 graphs A and B seem to have some printing mistake. Letter E for graph E is missing.

The following sentence is repeated in Page 6 of Discussion: “Individual manufacturers potentially could have selected the most optimal stabilizer combinations and freeze-dry conditions depending on the strain in the vaccine and as such have considerably improved upon robustness of the vaccine preparation.”

Sometimes in Discussion you refer to clone30 and Cl/79, or the other strains instead of vaccines A-F. It is hard to follow both nomenclatures. Can you say perhaps “clone 30 (Vaccine B), so the reader doesn’t have to go back to Table 1 and loose the track of what is being explained in the Results or in Discussion?

Author Response

In the abstract, could you consider rewriting the following sentence? This reviewer was unable to understand the idea on this sentence: “The data indicate preparation that specific factors may influence thermal stability rather than strain specific characteristics.” It can be unerstood only after finishing reading the manuscript.

We are sorry, there was a mix up in the order of words. Now it is: “These results point to preparation specific factors that influence thermal stability beyond strain specific characteristics.”

In my opinion the abstract lacks clarity at the end. By reading the abstract I cannot realize at its full extent what the conclusions are based on the main findings. Please revise it. Unlike the abstract, the Introduction provides more elements and clarity.

We recognize, that the conclusions were not clearly associated to the reported findings. We reorganized the abstract to point out in a more pronounced way our two key findings a) recognition of preparations specific factors influencing stability, and b) all ND vaccine strains were sensitive to inactivation by temperatures beyond 50°C.

Could you please confirm the residual titers in Page 5: “While at day 7 after temperature exposure infectivity was present in all other vaccine virus stocks, at the end of the experiment on day 10, residual infectivity was detectable only in viral stocks from vaccine D, E and A with viral-titer of 3.7x100, 1.4x102, and 1.6x102 TCID50 respectively”.

Thank you for pointing this out: We checked that and as shown in sTab2B the residual titer for vaccine was 1.58x10^2 TCID50/ml and for vaccine E 1.34x10^2 TCID50/ml. This appears in contradiction to the graph. However, we are showing residual infectivity in Fig 2F and original titer of vaccine E (day 0: 100%=2.08x10^8 TCID50) was lower than vaccine A (day 0: 100%=5.14x10^8 TCID50). Thus, even though the actual titer at day 10 is lower for vaccine E than for vaccine A the residual infectivity for vaccine E is higher. Considering the biological fluctuation of titration systems, these differences are marginal.

Figure 2 graphs A and B seem to have some printing mistake. Letter E for graph E is missing.

Thanks for pointing this out. The label “E” was lost during pasting it from sigma plot. The label has been added.

The following sentence is repeated in Page 6 of Discussion: “Individual manufacturers potentially could have selected the most optimal stabilizer combinations and freeze-dry conditions depending on the strain in the vaccine and as such have considerably improved upon robustness of the vaccine preparation.”

Thanks for pointing this out this embarrassing “left over” from the editing process. The first sentence has been removed.

Sometimes in Discussion you refer to clone30 and Cl/79, or the other strains instead of vaccines A-F. It is hard to follow both nomenclatures. Can you say perhaps “clone 30 (Vaccine B), so the reader doesn’t have to go back to Table 1 and loose the track of what is being explained in the Results or in Discussion?

This is a very valuable suggestion and we added the information.

Reviewer 2 Report

Osman et al test thermal stability of a number of commercial ND vaccines. This study is well carried out and clearly shows differences in thermal stability between various vaccines and highlights the importance of appropriate storage. I have no major issues with the study, which is useful for better understanding these parameters.

Minor comments:

  1. English language needs editing.
  2. Figures 2A and B: part of the images seem lost.
  3. Discussion could be made more concise.

Author Response

1.English language needs editing.

The manuscript has been edited thoroughly.

2.Figures 2A and B: part of the images seem lost.

A new version of Figure 2 has been added, including the missing label.

3.Discussion could be made more concise.

We recognized that some parts of the discussion was distracting form the main findings. We deleted the parts about side findings and hope that for the reader it became easier to keep the Focus.

Reviewer 3 Report

The presented data are sound. However, there are a few points that need to be addressed and modify in the manuscript before publishing: Major revision.

  1. Abstract and introduction should contain the scientific paucity of the previous studies that were addressed in this study.
  2. Keywords should be in alphabetical order.
  3. Does haemagglutination titer evaluation is sufficient to determine the efficacy of the vaccine stabilizers?
  4. Authors should check the conclusion part: Are the results really supporting the conclusion? The tested /selected parameters are sufficient for the conclusion?
  5. References should be cited by following journal-style/format.
  6. Need to check for typographical errors, plagiarism, punctuation, and grammar throughout the manuscript.

Author Response

1.Abstract and introduction should contain the scientific paucity of the previous studies that were addressed in this study.

See comments for reviewer 1. The abstract has been edited in order to emphasis the link between the findings and conclusions.

2.Keywords should be in alphabetical order.: done

3.Does haemagglutination titer evaluation is sufficient to determine the efficacy of the vaccine stabilizers?

Thanks for pointing this out: Our data clearly demonstrate, the HA titer remains stable even though infectivity dropped. We didn´t focus on that kind of virological obvious part, but included a respecitve indication in the discussion.

4.Authors should check the conclusion part: Are the results really supporting the conclusion? The tested /selected parameters are sufficient for the conclusion?

The conclusions that preparation-specific factors are influencing stability are supported by the results from testing lyophilized vs. dissolved original vaccine preparations. In addition testing of a separately prepared virus stocks giving divergent results are further strengthening the conclusion that preparation of a vaccines is influencing stability of a product. The factors that might influencing stability were discussed, but as we don´t have any data on that, did not draw any conclusion.

The second conclusion that all types of live ND-vaccines require a cold chain is derived from the experiments with elevated temperatures. In particular in light of the discussions on vaccine failure, we wanted to highlight the importance of sensitivity to temperature of above 40° C. This should be considered not only for the transport of vaccines, but also for the application of a vaccine and we would be happy if this publication will help to strengthen this point.

5.References should be cited by following journal-style/format.

The style of the references has been edited.

6.Need to check for typographical errors, plagiarism, punctuation, and grammar throughout the manuscript.

The manuscript has been edited thoroughly.

Round 2

Reviewer 3 Report

The presented data are sound. However, there are a few points that need to be addressed and modify in the manuscript before publishing

  1. Once again check the abstract and introduction section for scientific paucity of the previous studies that were addressed in this study.
  2. The authors should check once again the tested /selected parameters are sufficient for the conclusion?
  3. Check once again the references to cite them following journal-style/format.
  4. Need to check once again for typographical errors, plagiarism, punctuation, and grammar throughout the manuscript.

Author Response

  1. Once again check the abstract and introduction section for scientific paucity of the previous studies that were addressed in this study.

The reviewer is right. Our study was not designed to verify results of specific studies and thus we did not address those studies in the abstract and/or introduction. However, we understand that the manuscript would benefit from a short statement on our motivation for this study. Considering the limitation of the length of an abstract, we confined this to a short complementation in the abstract and introduction (see red sentence below).

Abstract: Vaccination against Newcastle disease (ND), a devastating viral disease of chicken, is often hampered by thermal inactivation of the live vaccines, in particular in tropical and hot climate conditions. In the past “thermostable” vaccine strains (I-2) were proposed to overcome this problem but previous comparative studies did not include formulation-specific factors of commercial vaccines. In the current study, we aimed to verify superior thermal stability of commercially formulated I-2 strains by comparing the thermal stability of 6 commercially available ND vaccines.

Introduction: What determines the stability of a certain vaccine might depend on intrinsic properties of the strain, but other factors such as manufacturing and lyophilization processes can have an equally important impact. Whereas studies on properties of specific strains in a standardized formulation are available [29; 30, 31]; data on the influence for thermostability of the galenics of ND vaccines are missing so far.

2.The authors should check once again the tested /selected parameters are sufficient for the conclusion?

We think that the two most relevant conclusions are well supported by our findings.

“…thermostability of a given vaccine does not depend solely on the used vaccine strain, but that the overall excellence of the manufacturing process can be as or even more important for the quality of the vaccine.”

This conclusion is supported by findings depicted in Fig 2A/ 2B, stability of original vaccines at 37° C (details see s Tab 2B). The results demonstrate that infectivity of two vaccines (vaccine D and F) containing an I-2 strain, are less stable than others containing regular vaccines (vaccine B and C). In addition, the other two vaccines containing an I-2 strain (vaccine A and E) did not outperform the regular vaccines. On the other hand, repetition of the experiments with diluted vaccines (Fig 2C / 2D) and experiments with at the laboratory and composed only of amnion-allantoic fluid (AAF) (Fig 2F) diminishes differences observed with lyophilized pellets. However comparing those two experiments we do observe, differences between virus from vaccines and fresh propagated virus stocks: clone 30 from dissolved vaccine is as stable as I-2 from vaccines but more labile when prepared virus stocks were tested. Contrary Cl/79 from dissolved vaccine is more sensitive compared to other vaccine strains, but comparably stable when prepared virus stocks were tested.

Specific factors causing this observed differences remain unclear. However, the above experiments in conjunction with our controls, testing stability at 4° C and testing samples at the beginning of the experiment, point to a factor intrinsic to these freeze dried products. This has been circumscribed as “overall excellence of the manufacturing process” and some factors are considered in the discussion (page 7, second paragraph).

Second major conclusion within the last paragraph of the discussion (page 9) has been modified:

“Our study clearly shows, that regardless of the ND strain used, the appropriate cold chain is mandatory for live ND vaccines.”

New: “From the practical point of view, the rapid decay of infectivity at 51° C and 61° C indicates that all vaccines tested are prone to inactivation at elevated temperatures. …… In order to ensure the application of an appropriate vaccine dose, maintaining the cold chain for live ND vaccines is necessary.”

We demonstrated for four freeze-dried preparation a comparable decay at 37° C, over the observation period of 21 days. This included two regular vaccines and two vaccines containing an I-2 strain (see above). At the same temperature but tested as dissolved vaccines or as propagated virus stock, infectivity is less stable but is retained for days (Fig.2 C-F). In contrast, when propagated virus stock were tested at elevated temperatures of 51° C and 61° C, infectivity is vanished within 4 to 2 hours respectively (Fig.3C-D). Testing vaccine B as freeze dried pellet, infectivity was more stable, but we find a sharp decrease by day one, resulting in virus titer of 1/10th (51° C) and 1/10,000 (61° C) of the aspired vaccine dose (Fig. 3A). It should be kept in mind that vaccine B (clone30) performed as god as I-2 strains when tested as freeze-dried pellet or dissolved from the product at 37° C (Fig. 2). Thus the observed rapid decay of all dissolved propagated virus stocks at 51° C and 61° C, indicates that all vaccines tested are prone to inactivation at elevated temperatures.

3.Check once again the references to cite them following journal-style/format.

Done.

4.Need to check once again for typographical errors, plagiarism, punctuation, and grammar throughout the manuscript.

The manuscript was edited by a colleague with certified translation skills who corrected typographical errors, punctuation and grammar. Plagiarism can be excluded by authors, all statements made, if not supported by own data, are accompanied by a citation of the relevant literature.

This manuscript is a resubmission of an earlier submission. The following is a list of the peer review reports and author responses from that submission.